Large-scale differences in diversity and functional adaptations of prokaryotic communities from conserved and anthropogenically impacted mangrove sediments in a tropical estuary

De Santana Carolina O. cal.uefsbio@yahoo.com.br 1
Spealman Pieter 2
Melo Vania 3
Gresham David 2
de Jesus Taise 4
Oliveira Eddy 4
Chinalia Fabio Alexandre 1
1 Geosciences Institute, Federal University of Bahia , Salvador , Bahia , Brazil
2 Department of Biology, New York University , New York City , NY , United States of America
3 Department of Biology, Federal University of Ceará , Fortaleza , Ceará , Brazil
4 Department of Biology, State University of Feira de Santana , Feira de Santana , Bahia , Brazil
Kormas Konstantinos
Electronic publication date: 2021 Sep 23
Publication date: 2021
Volume: 9
Electronic Location ID: e12229
Received 2021 Apr 13; Accepted 2021 Sep 8
Copyright: ©2021 De Santana et al.
Copyright year: 2021
Copyright holder: De Santana et al.
License: This is an open access article distributed under the terms of the Creative Commons Attribution License, which permits unrestricted use, distribution, reproduction and adaptation in any medium and for any purpose provided that it is properly attributed. For attribution, the original author(s), title, publication source (PeerJ) and either DOI or URL of the article must be cited.
License URL: https://creativecommons.org/licenses/by/4.0/

Keywords: Mangrove, Urbanization, Sediment microbiome, Environmental impact, Tropical estuary

Funding: Coordenação de Aperfeiçoamento de Pessoal de Nível Superior – Brasil (CAPES) Finance Code 001 Fundação de Amparo à Pesquisa do Estado da Bahia – Fapesb 19.571.128.2566 This work was financed by the Coordenação de Aperfeiçoamento de Pessoal de Nível Superior – Brasil (CAPES) – Finance Code 001 and by the Fundação de Amparo à Pesquisa do Estado da Bahia – Fapesb (No.19.571.128.2566). There was no additional external funding received for this study. The funders had no role in study design, data collection and analysis, decision to publish, or preparation of the manuscript.

==============================
Mangroves are tropical ecosystems with strategic importance for climate change mitigation on local and global scales. They are also under considerable threat due to fragmentation degradation and urbanization. However, a complete understanding of how anthropogenic actions can affect microbial biodiversity and functional adaptations is still lacking. In this study, we carried out 16S rRNA gene sequencing analysis using sediment samples from two distinct mangrove areas located within the Serinhaém Estuary, Brazil. The first sampling area was located around the urban area of Ituberá, impacted by domestic sewage and urban runoff, while the second was an environmentally conserved site. Our results show significant changes in the structure of the communities between impacted and conserved sites. Biodiversity, along with functional potentials for the cycling of carbon, nitrogen, phosphorus and sulfur, were significantly increased in the urban area. We found that the environmental factors of organic matter, temperature and copper were significantly correlated with the observed shifts in the communities. Contributions of specific taxa to the functional potentials were negatively correlated with biodiversity, such that fewer numbers of taxa in the conserved area contributed to the majority of the metabolic potential. The results suggest that the contamination by urban runoff may have generated a different environment that led to the extinction of some taxa observed at the conserved site. In their place we found that the impacted site is enriched in prokaryotic families that are known human and animal pathogens, a clear negative effect of the urbanization process.

Introduction

Mangrove forests are coastal ecosystems in tropical and subtropical areas, accounting for more than 137,760 km2 of the world’s coastline, with one of the largest areas being the Brazilian coast (Spalding, Kainuma & Collins, 2010; Giri et al., 2011). These ecosystems are recognized to be of strategic importance for climate change mitigation, due to their large capacity for carbon sequestration and storage, as well as for protecting the coast from erosion and rising sea levels (Howard et al., 2017; Macreadie et al., 2019). Many aspects of mangrove health and function are tied to microbial metabolic activities that play essential roles in large-scale biogeochemical nutrient cycling and as such have been the subject of many genomic studies (Yun, Deng & Zhang, 2017; Zhou et al., 2017; Imchen et al., 2017; Imchen et al., 2018; Marcos et al., 2018; Lin et al., 2019; Allard et al., 2020).

Globally, mangrove ecosystems are threatened by habitat degradation and loss due to anthropogenic disturbances such as urbanization, industrial development and increasing population densities in coastal areas (Fernandes et al., 2014; Imchen et al., 2018; Gong et al., 2019; Trevathan-Tackett et al., 2019). In Brazil, where a large number of mangrove studies have been performed, numerous studies have been conducted in disturbed mangroves (Andreote et al., 2012; Sanders et al., 2014; Cabral et al., 2016), mostly without direct comparison to undisturbed sites. Undisturbed mangrove sites have been conserved within Brazil’s Permanent Protection Areas, as defined by the National Council on the Environment (CONAMA) through the resolution No. 303 of 2002 (CONAMA, 2002). Critically, this resolution was revoked in September of 2020, thus increasing the risks of degradation of such areas in the future.

One important Permanent Protection Area in Brazil, located in Bahia State, is the Pratigi area which is characterized by the presence of small urban areas interspersed within dense Atlantic Rainforest vegetation. Notably, when the Pratigi Protection Area was created in 1998, several small urban locations were already present, and were integrated in the efforts for the sustainable use of the natural resources (MMA, 2004). Despite the presence of these urban assemblages, the area has continuously received high environmental quality indices (Lopes, 2011; Ditt et al., 2013; Mascarenhas et al., 2019; Carneiro et al., 2021). Nevertheless, even with this high level of environmental quality across the entire area, some local anthropogenic disturbance is still present, such as construction in what was a mangrove forest around the city of Ituberá (Fig. 1). In this location, it is possible to observe clear signs of anthropogenic impact, such as raw sewage and urban runoff, as well as the withdrawal of the native vegetation from the immediate vicinity.

Figure 1 Location map of the collection sites of the pristine and impacted mangrove areas.

The existence of a small locus of disturbance within a largely conserved estuarine system provides for an opportunity to study how human activities impact microbial populations. In this study, we aimed to assess the impacts of urbanization on the surrounding mangrove area by examining the prokaryotic communities of sediments and their potential functional roles in the context of the local disturbance around Ituberá city, in comparison to the results of a distant and conserved spot in the same estuarine system. Considering the increased risks of environmental disruption now that the protection legislation has been revoked, this study provides important elements for the understanding of the consequences of even light urbanization and minor human interference in a mangrove area.

Prokaryotes, specifically bacteria, comprise a majority of the sediment microbiomes of mangrove ecosystems in Brazil and worldwide, accounting for 94.8 to 99.2% of the microbial diversity (Andreote et al., 2012; Imchen et al., 2018). Previous research conducted in Goa, India (Fernandes et al., 2014) and the Red Sea region (Ullah et al., 2017) found higher diversity in anthropogenically impacted mangroves in comparison to conserved areas, possibly due to the higher availability of a wide variety of organic and inorganic compounds, which favors the colonization of exogenous opportunistic prokaryotes; ultimately creating significantly distinct communities. In this study, we hypothesized that proximity to urban environments and sources of constant input of domestic runoff leads to large-scale changes in prokaryotic composition and functional profiles in the microbiome of mangrove sediments. Considering the importance of prokaryotes in such areas, we conducted 16S rRNA gene sequencing analysis in both conserved and impacted mangrove areas within the Serinhaém estuary, and measured the correlations between the prokaryotic communities and environmental factors for each site. Using the taxonomic abundances obtained with the analysis of the 16S rRNA, we estimated the general patterns of functional potentials in these microbiomes.

Materials & Methods

Study area

The Serinhaém estuary (Fig. 1) is located within the limits of the Environmental Protection Area of Pratigi in the state of Bahia, Brazil. The Pratigi Protection Area is defined as an area of sustainable use aiming for the protection of remnants of the Brazilian Atlantic Forest and associated ecosystems, such as restingas and mangroves, in which extractive and agricultural activities are still allowed but regulated (MMA, 2004; Ribeiro et al., 2019). This estuarine region covers approximately 32 km of the Juliana River basin, a riverine system that is completely inside the limits of the protected area, and flows into Camamu Bay, a tropical oligotrophic estuarine system, where it meets the Atlantic Ocean. Ituberá is a relatively small urban area of approximately 28,000 people (IBGE, 2020) within the Protection Area, with a strong economic focus on tourism. For the selection of collection sites we coordinated with the Organização de Conservação da Terra (OCT) which manages conservation actions within the protection area of Pratigi.

Sampling

The two collection sites were located 9.5 kilometers distant from each other. The mangrove trees at both collection sites are composed of the species Rhizophora mangle, Avicennia schaueriana and Laguncularia racemosa (MDMA, 2010). Collections were performed at morning low tide. For both sites, cores consisting of the top 10 cm of the surface layer were collected using a stainless-steel cylindrical sediment core sampler with a diameter of 10 cm, always avoiding mangrove trees and rhizosphere associated sediments (De Santana et al., 2021).

At the conserved mangrove site (13°44′34.6″S, 39°03′30.6″W), collection points withing each tidal zone (supralittoral, intertidal and sublittoral) were 15 m distant from each other. Within each tidal zone we collected three sediment samples, each sample consisting of three combined sediment cores taken within an immediate vicinity (50–60 cm), resulting in nine samples at this site (De Santana et al., 2021). Collections were made in July 2018. The conserved collection site exhibited no visible signs of anthropogenic disturbance or pollution. For this study, we recombined the FASTQ files of these nine samples into three composite replicates, where each replicate included one sample from each tidal zone (Text S1).

The samples from the impacted site (13°44′11.1″S, 39°08′46.0″W) located just outside the city of Ituberá, were collected in triplicate, each sample was located 5 m from the other two in a triangle covering the sublittoral zone. As with the conserved samples, each impacted sample was composed of three sediment cores, taken within an immediate vicinity of 50–60 cm. Collections were made in February 2020. The definition of mangrove associated tidal zones in this area was complicated by the advance of the city’s construction, resulting in a smaller and partially deforested area. The emerged sediments at the time of collection showed pollution by domestic runoff and construction leftovers. Because of the absence of waste management infrastructure, numerous developments along the shore where the sediments were taken exhibited sewage waste pipes directly releasing into the environment (Fig. S1). In both collection sites the sediments from the submerged zone present a silt/mud constitution (Santos & Nolasco, 2017) and were collected within the tree line.

Sediments were transported in plastic bags inside thermic boxes filled with ice to the laboratory of Petroleum Studies in the Federal University of Bahia. Samples from both sites had plants and other macroscopic organic materials removed before subsequent procedures.

For each sediment sampling site we measured physical-chemical parameters such as dissolved oxygen, conductivity, pH and temperature in the water column directly above the sampling spot of submerged sediments, using a multiparameter monitoring system (YSI model 85, Columbus). For each sediment core an aliquot was separated for the measurement of organic matter content using the ‘loss-on-ignition’ method (Nelson & Sommers, 1996) and the other part was kept in the −20 °C freezer for DNA extraction.

For heavy metal analysis we relied on previously performed work (Pereira, 2016) which identified the background metal concentrations in the sediments of the entire estuarine channel. The analysis of heavy metals was carried out as described in Pereira (2016). Briefly, 0.5 g of the clay fraction of the air-dried samples were transferred to test tubes (25 × 25 mm) added with a solution of hydrochloric acid (HCl) and nitric acid (HNO 3) at the ratio 3:1 (three mL HCl and one mL HNO3) and placed in a 100 °C digester block for 24 h along with analytical blanks. After digestion, samples were filtered and placed in 50 ml volumetric flasks, added with ultrapure water to the extent of 25 ml and subsequently analyzed to determine the concentrations of the metals Al, As, Ba, Co, Cr, Cu, Fe, Li, Mn, Ni, Pb, Sn, V, and Zn by inductively coupled plasma optical emission spectrometry (ICP / OES) (Agilent Technologies 700 series). Data from those downstream sites closest to the sediment sampling sites (Text S1), had an average distance of 1.64 km for the impacted site and 6.98 km for the conserved site. The Kruskal-Wallis test was performed to identify significant differences in values between the two sites (Text S1).

DNA extraction, amplicon library generation, and sequencing

Total genomic DNA was extracted using PowerSoil DNA Isolation Kit (Qiagen, Carlsbad, CA, USA) from 0.25 g of each composite sample and stored at −80 °C before amplification, resulting in 9 DNA samples for the conserved site and 3 DNA samples for the impacted site. For amplification of the V4 region of the prokaryotic 16S rRNA gene we performed PCR using the following primer pairs. For the conserved site we used 515F-Y (Parada, Needham & Fuhrman, 2016) and 806R-XT (Caporaso et al., 2011), while 515F and 806R (Kozich et al., 2013) were used for the impacted samples. Each sample required a minimum of 12.5 ng before PCR as quantified using Qubit (Thermo Scientific, Waltham, MA USA). For PCR we used the following protocol: for 2.5 µl of each sample (minimum 5 ng/L), 5 µl of the forward and reverse primers and 12.5 µl of the 2 × KAPA HiFi HotStart ReadyMix were added, to the total volume of 25 µl. The samples were then subjected to following cycles: 1 × 95 °C for 3 min, 25 × 95 °C for 30 s, 1 × 55 °C for 30 s and 1 × 72 °C for 30 s and 1 × 72 °C for 5 min. Negative controls were also used in parallel and ran on an agarose gel, as no band resolved these were not sequenced. PCR cleanup was performed using Ampure XP beads. Amplicon libraries were prepared using the Nextera XT according to manufacturer’s directions (Illumina, San Diego, CA, USA). Final quantification and pooling were performed using KAPA HiFi. For the conserved site we loaded ∼6 pM of each sample for paired-end sequencing (2 × 150) (Caporaso et al., 2012) performed using the Illumina MiSeq platform (Illumina, San Diego, CA, USA), V2 kit (300 cycles), while the impacted site we used ∼48 pM of each sample for paired-end sequencing (2 × 250) using the Illumina NovaSeq XP (Illumina, San Diego, CA, USA). It is possible that these methodological differences in library preparation and sequencing may produce artifactual differences in the two data sets.

Data analysis

Read preprocessing

Demultiplexed sequences were filtered and trimmed with Trimmomatic (Bolger, Lohse & Usadel, 2014) (ILLUMINACLIP:TruSeq3-PE.fa:2:30:10 LEADING:3 TRAILING:3 SLIDINGWINDOW:4:15 MINLEN:100) with the requirement of a minimum average read quality score of 15 for inclusion. For each read, the sliding window cuts any read at the point where the median quality score over a 4nt window is less than 15.

Paired-end merge, denoise, and chimera removal using DADA2

Paired reads from the conserved site were combined as described previously (De Santana et al., 2021), briefly, paired reads were combined using QIIME2 for reads with an overlap greater than 9 while paired reads with an overlap greater than 6 were combined using a custom script (Article S1) before denoising using DADA2 (denoise-single, –p-trim-left 3, –p-trunc-len 0, –p-max-ee 2.0, –p-trunc-q 2). The reads from the impacted site were also denoised using DADA2 (denoise-paired, –p-trim-left-f 13, –p-trim-left-r 13, –p-trunc-len-f 150, –p-trunc-len-r 150). These were then merged with the denoised sequences from the conserved site for further analysis. Reads were resolved and clustered into amplicon sequence variants (ASVs) using QIIME2 (Bolyen et al., 2019). All ASVs were retained in the data set. Filtering was performed only on taxa and only for differential abundance analysis.

Taxonomic assignment was performed using QIIME2’S naive Bayes scikit-learn classifier (Bokulich et al., 2018) trained using the 16S rRNA gene sequences in SILVA database (Silva SSU 132), (McDonald et al., 2012), (Text S1). Taxonomic counts were also used for hierarchical correlation clustering (Text S1) tests using gneiss in QIIME2.

Unnormalized ASV (Table S1) and taxonomic abundances (Table S2) as well as overlap (Text S1) between sites are reported in Supplemental Information.

ASV analysis using QIIME2

Alpha-rarefaction curves were generated using QIIME2 (Text S1). We performed a variety of alpha-diversity (Kruskal–Wallis statistic, Fig. S2) and beta-diversity tests (PERMANOVA statistic, Fig. S3). As both Kruskal–Wallis and PERMANOVA are non-parametric tests we did not test for normality. However, both are sensitive to homoscedasticity. For Kruskal–Wallis we used Levene’s test (Df = 1, F-value = 0.2275, p-value = 0.6334) and betadisper (Anderson, 2006) for PERMANOVA (Df =1, Sum_Sq = 0.002209, Mean_Sq = 0.002209, F-value = 0.8045, p-value = 0.4205).

Environmental variable correlations using Vegan

Correlations between taxonomic community structure and environmental variables associated with the sample sites were tested using the Vegan package (version 2.5-6) (Dixon, 2003) in R (R Core Team, 2019). Distance matrices were calculated using vegfit (default settings) on taxa abundance identified by QIIME2 for each site. We performed constrained ordination of the distance matrices (PCoA) using envfit (default settings).

Potential functional analysis using PICRUSt2

We used PICRUSt2 (version 2.3.0-b) (Ye & Doak, 2009; Louca & Doebeli, 2018; Douglas et al., 2019; Barbera et al., 2019; Czech, Barbera & Stamatakis, 2020) with default settings for functional analysis using the observed ASV abundances generated by QIIME2. It is important to note that functional predictions generated by PICRUSt2 using 16S rRNA sequences alone will not be as accurate as a metagenome including the functional sequences themselves would be (Sun, Jones & Fodor, 2020). Because we are reliant on 16S rRNA gene all KEGG ortholog (KO) abundances are derived from the closest reference taxa that matches the supplied taxa. As such there may be significant differences between the KO of the actual organism and the reference we rely on.

To address this, we use PICRUSt2’s nearest sequenced taxon index (NSTI) as a heuristic cutoff in the evaluation of taxon level functional analysis (Table S3). Although we are using taxonomic families in this study, we will continue to use the NSTI cutoff of 0.15, despite it being derived for species level comparisons. Where relevant, families with median NSTI scores within 1 standard deviation of 0.15 are labeled with an asterisk (*). Only one family, Pirellucae, was both significantly enriched in metabolic KOs while also having an NSTI outside of this range; this family was not included in the analysis.

KEGG Orthologs (KOs) were subsequently analyzed for significant (p-value ≤0.05) differential abundances after centered log-ratio transformation (aldex.clr) using the general-linear model method (aldex.kw) of the ALDEx2 package (ver 1.18.0). These were then used for the construction of the heatmaps of KOs with differential abundance between the conserved and impacted mangrove. Pathway analysis of these KOs was performed using KEGG Mapper (Kanehisa & Sato, 2020). For a pathway to be considered we required that the entire pathway needed to be significantly enriched at a single site (i.e., ‘complete’).

Differential abundance analysis

In order to identify which taxa were significantly different in abundance in each sampling area we carried out a taxa enrichment analysis. ASV abundances were normalized by total sum scaling (TSS) wherein each ASV read abundance is reduced by a fraction so that the total sample is downsampled to match the least abundant sample in the experiment (Weiss et al., 2017). ASVs were combined into assigned taxa using the taxonomic results of QIIME2. To be defined as significantly different, several criteria must be met: there must be at least 100 unnormalized counts of the taxa in at least one site, the sets of observations between sites must be statistically significant (p-val ≤0.05) as calculated using the Kruskal–Wallis H test and finally, the effect size must be in excess of a 20% difference in abundance between sites.

Since TSS normalization can confound interpretation, we also applied the same method to data that had been normalized using cumulative sum scaling (CSS). CSS is a normalization approach that weights the normalization factor based on the relative abundance of ASVs against the total abundance (Paulson et al., 2013). However, as we observed that this approach resulted in the loss of several common taxa and the gain of many more marginal taxa (Text S1) we chose to use the TSS normalization method.

A further extension of differential abundance is to identify taxa which are exclusive to a single site. In order to determine taxa exclusive to a single site with a high degree of certainty we required a taxon to have been observed 0 times (using unnormalized abundances) at one site and at least 100 counts at the other, with a minimum of 10 counts per replicate.

Gneiss hierarchical clustering is an alternative approach that uses balance calculations to extend differential abundance analysis beyond species and instead identify niche specific subcommunities (Morton et al., 2017). Using this approach hundreds of taxa were found to be differentially abundant between the two sites (Text S1) with all but 3 of our predicted differentially abundant taxa being present in the appropriate enriched cluster.

Taxa specific predicted contribution to metabolic pathway functional abundances

We calculated metabolic pathway enrichment specific to a given taxa at a given taxonomic level. For this we relied on the predicted relative functional abundance results of PICRUSt2. First, for each taxon, at each level, we combined all predicted relative functional abundances of KOs belonging to certain metabolic pathways: methane (ko00680), nitrogen (ko00910), sulfur (ko00920), and pentose phosphate (ko00030) metabolism - as well as, carbon fixation pathways in prokaryote (ko00720), carbon fixation pathways in photosynthetic organisms (ko00710), and photosynthesis (ko00195). We then applied Kruskal–Wallis H test to identify taxa that had significantly different taxa-specific pathway abundances. Furthermore, we required that a taxon at a single site must be significantly enriched relative to the other sites and that the taxa contributes at least 5% of the total relative functional abundance in at least one site.

The entire computational workflow is available as a repository in GitHub: https://github.com/pspealman/Project_Impact.

The impacted mangrove sediment sequencing data is available from NCBI BioProject PRJNA650560, while the pristine mangrove sediment data is available from NCBI BioProject as accession number PRJNA608697.

Results

Structural aspects of prokaryotic communities of mangrove sediments

Taxonomic assignment of the prokaryotes in the mangrove samples resulted in a total of 7,278 ASVs (Table S1), belonging to 1,369 taxa (Table S2), 861 of which had at least 20 unnormalized reads in at least one site (Text S1). These resolved to 184 taxa at the family level or higher, with 124 having at least 20 unnormalized reads at a single site. 60 (48%) of these were present at both sites, while 13 (11%) were unique to the conserved site, and 51 (41%) were unique to the impacted site (Text S1). Bacteria accounted for nearly 91% of all reads while archaeal taxa accounted for 7.9% and unassigned groups only 1.6%. The two mangrove areas presented considerable differences at the phylum level for the 12 most abundant phyla. Figure 2 displays all the classes belonging to the top 12 phyla with more than 1% abundance in the analysis. Some phyla such as Firmicutes and Planctomycetes presented a large decrease in abundance in the impacted site when compared to the conserved site, while the presence of the entire phylum Euryarchaeota was only detected in the urbanized mangrove (Fig. 2). Some phyla were represented by different classes in each site, as is the case of Chloroflexi, with Dehalococcoidia only prevailing in the impacted mangrove site. Below the family level we observed a dominance of unassigned groups (Text S1).

Figure 2 Taxonomic abundances of prokaryotes in each mangrove site.

Summed normalized read abundances from across replicates of prokaryotic groups for phylum and class taxonomic levels at each mangrove site.

The analysis revealed significantly higher richness (Total ASVs, Kruskal–Wallis H = 3.857, df = 1, p-value = 0.050) and diversity (Shannon’s diversity, Kruskal–Wallis H = 3.857, df = 1, p-value = 0.050) in the impacted mangrove sediments (Figs. 3A and 3B), relative to the conserved sediments. Additional alpha-diversity test results are available in Text S1. Prokaryotic community composition differed between the two sites with marginal significance (PERMANOVA, permutations = 999, pseudo-F = 3.285, p-value = 0.096), (see Text S1 for additional beta-diversity tests) (Fig. 3C). Samples of each mangrove site form separate clusters as can be seen in the PCoA plot (Fig. 3D). The PCoA plot also shows that the samples from the conserved mangrove site present higher variability than the samples collected at the impacted site. The results showed a significant (p-value ≤0.05) correlation between the structure of prokaryotic communities and some environmental variables (Text S1). These include organic matter (conserved site mean: 3.8%, impacted site mean: 9.9%), temperature (27.7 °C, 29.3 °C), and Cu (0 mg Kg−1, 0.16 mg Kg−1). While dissolved oxygen (5.54 mg L−1, 8.21 mg L−1), pH (7.62, 7.46), Ba (0.66 mg Kg−1, 0.44 mg Kg−1), and salinity (15.1 ppt, 13.3 ppt) were only marginally significant (p-value ≤0.1) (Fig. 3D).

Figure 3 Prokaryotic diversity and environmental factors interacting to create distinct communities.

Here we show total ASVs per site (A), alpha-diversity using Shannon’s Index (B), and beta-diversity using Bray–Curtis (C). A PCoA plot of the sample sites with fitted vectors of environmental variables (D) shows that organic matter, temperature, and Cu are significantly correlated with the prokaryotic communities, while dissolved oxygen, pH, Ba, and salinity are marginally significant.

The majority of families with significant (p-value ≤ 0.05) differences in abundances between sites were found in the sediments of the impacted mangrove. Figure 4 shows the 64 families with significant and large effect size differences between the sites (>20%) and those that are absent in one of the areas. We found 35 families that had significantly higher abundance in the impacted mangrove site, including human pathogen associated families Burkholderiaceae (Coenye, 2014), Pasteurellaceae, Spirochaetaceae, Ruminococcaceae, Veillonellaceae, and Rikenellaceae. Also more prevalent in the impacted sediment were known archaeal methanogens Methanomicrobia and Thermoplasmata (Whitman, Bowen & Boone, 2014), Fe and Mn metabolizing Geobacteraceae (Röling, 2014), and members of Gammaproteobacteria associated with sulfur metabolism Syntrophobacteraceae (Liu & Conrad, 2017), Thioalkalispiraceae (Mori & Suzuki, 2014), Saccharospirillaceae (Vavourakis et al., 2019), Thiomicrospiraceae (Eberhard, Wirsen & Jannasch, 1995); methanogen Methylomonaceae and nitrogen metabolising Nitrincolaceae. We also found Lachnospiraceae and Anaerolineae members, which were found to be of greater abundance in PVC contaminated environments (Seeley et al., 2020). We also found 9 families with higher abundance in the conserved sediments. These include Thermoanaerobaculaceae, recently shown to have a high metabolic diversity in wastewater treatment plants (Kristensen et al., 2021), which along with Pirellulaceae, was recently found to be sensitive to microplastic pollution (Seeley et al., 2020). All enriched members of Planctomycetes, some of which are capable of anaerobic ammonia oxidation (anammox) have greater abundance at the conserved site, except MSBL9 SG8-4, potentially because of its freshwater preference (Andrei et al., 2019). We also observed the pathogen associated Vibrionaceae (Farmer, 2006) in the area.

Figure 4 Taxa with significant differences between impacted and conserved mangrove sites.

Prokaryotic families with significant differences in abundances and large effect size differences (>20%) between the mangrove sites. Some families show zero abundance in one of the studied areas.

Functional aspects of prokaryotic communities of mangrove sediment

Functional profiles of the prokaryotic communities generated with PICRUSt2 based on the taxonomic abundances allowed us to identify statistically different potentials for carbon, nitrogen, phosphorus and sulfur metabolic pathways between the conserved and impacted mangrove sites (Figs. 5 and 6).

Figure 5 Carbon metabolism pathways and different abundances between sites.

Abundance heatmaps of carbon metabolic pathways in each site. (A) Carbon fixation for photosynthetic organisms. (B) carbon fixation for not photosynthetic organisms. (C) methane metabolism. (D) photosynthesis.

Figure 6 Abundances of metabolic pathways of different elements at each site.

Heatmaps of metabolic pathways abundances for nitrogen (A), sulfur (B), pentose phosphate (C), photosynthesis (D).

The carbon metabolism heatmaps are subdivided in: carbon fixation for photosynthetic organisms (ko00710), prokaryotes (ko00720) and methane (ko00680) metabolism, in order to allow a better visualization (Fig. 5). Generally, the methane and carbon fixation pathways presented higher functional abundance in the impacted site than that of the conserved mangrove. We found that the impacted site is enriched in the formaldehyde assimilation pathway (M00345), methane metabolism modules, (M00563) methanogenesis, 2-oxocarboxylic acid chain extension (M00608), light Crassulacean acid metabolism carbon fixation (M00169), and serine biosynthesis (M00020).

The nitrogen metabolism heatmap (Fig. 6) presents enrichment of metabolic pathways in the impacted mangrove site, with a significant enrichment in the nitrogen to ammonia fixation module (M00175). The functional profile of pentose phosphate metabolism also shows a tendency for enrichment of KOs in the impacted mangrove area. This enrichment includes the pentose phosphate pathway (M00004), PRPP biosynthesis (M00005), both oxidative (M00006) and non-oxidative phases (M00007), the Entner-Doudoroff pathway (M00008), as well as the archaeal pathway (M00580). For sulfur metabolism, we found that some KOs are absent or nearly absent in the conserved sediments while generally present in the impacted area. In the conserved area, we found significant enrichment only in the cysteine biosynthesis (M00021) pathway. While photosynthesis associated KOs were relatively low, we did find significant enrichment in the conserved sediment for both photosystems I (M00163) and II (M00161), we also found that the impacted site is significantly enriched in F-type ATPase ATP synthesis (M00157). Taken together, these findings suggest that the high abundance and diversity in the impacted area leads to enrichment in several metabolic pathways, including several wholly absent from the conserved area.

We also analyzed the contribution each family makes to the metabolism of the elements C, N, P and S. Families that contribute 5% or more to a given nutrient cycle in at least one site are represented in Fig. 7. We found the majority of the families with large contributions to the metabolic pathways in the conserved mangrove sediments, which display the lower biodiversity. The families Bacillaceae and Stappiaceae were greatly important for nutrient metabolism in the conserved mangrove sediments but not in the impacted sediments. Conversely, only Syntrophaceae, and Desulfarculaceae presented large functional contributions for the impacted mangrove area. Anaerolineaceae and Desulfobacteraceae showed important contributions for the functional potentials in both studied areas. This is consistent with a lower biodiversity leading to a higher number of taxa making larger contributions to the total functional potential.

Figure 7 Taxa responsible for the metabolic potentials.

Relative contributions of the prokaryotic families that contribute for >5% of the metabolic potentials in at least one site.

Discussion

Anthropogenic impacts are known to cause important changes both in structure and function of sediment microbiomes in mangroves (Ottoni et al., 2017; Imchen et al., 2018; Allard et al., 2020). In our study, we observed biodiversity to be higher in the impacted mangrove sediments than at the conserved site. Similar observations have been made previously in the comparison between conserved and anthropized mangrove areas in India (Fernandes et al., 2014) and the Red Sea region (Ullah et al., 2017). Here we find that organic matter is one of the most significant drivers of prokaryotic structuring between the measured environmental variables (Fig. 3). The large differences in organic matter contents could be the product of the urban runoff at the impacted site, which presented higher levels (median OM 9.9%) in comparison with the conserved area (median OM 3.8%). This suggests that this environment may have become more eutrophic due to the constant exposure to urban tailings, as observed in locu, which results in greater biomass production and biodiversity (Fernandes et al., 2014; Glibert, 2017). While some studies that use incubation or microcosm approaches have observed decreasing microbial diversity as a result of nutrient addition, in experiments with coastal soils (Aoyagi et al., 2015; Wang, Huang & Zheng, 2016; Bulseco et al., 2019; Craig et al., 2021), our study relied on field sediments which are continually exposed to anthropogenic presence, as such, would be expected to differ from controlled microcosms.

Besides organic matter, the structure of the prokaryotic communities in these mangrove sediments was greatly influenced by other environmental factors, such as temperature, copper content, pH, salinity and barium concentrations (Fig. 3D). Some of these, such as salinity and pH are the result of naturally driven biogeochemical processes in the area, while others, such as metal content, were observed to reflect the natural lithology of the estuary area (Pereira, 2016) or tidal deposition, as is the case with Ba (Carneiro et al., 2021).

Beyond the presence of an urban complex, other differences exist between the two sites as they are located along a gradient from river source to ocean. Notably, the increasing salinity due to proximity to the sea can impair the bioavailability of organic matter in these sediments, consequently affecting the availability of nutrients (Filippino et al., 2011). The different sources of dissolved and particulate OM in aquatic environments also have large effects on the contents, species, and availability of nutrients such as N, P, and heavy metals (Dong et al., 2020). Although mangrove species show an optimal growth over a large range of salinity (Krauss et al., 2008), and the differences observed in this study are small, all these factors may interact synergistically with the observed anthropogenic impacts, such as deforestation and construction byproducts, as well as urban tailings, to create a completely different sediment environment.

We sought to identify differences in the metabolic systems of the two sites using functional profiling. However, the functional profiles were not obtained by metagenomic sequencing, but based on the abundances of identified taxa compared to a reference holotype. To highlight this important limitation, we refer to these functional profiles as potential metabolic abundances. Although the use of metagenomic sequencing can provide better resolution to functional profiling (Zhang et al., 2021), it is possible to recover the general patterns of the microbiome functions through the taxonomic abundances of the 16S rRNA genes (Jovel et al., 2016). The diversity of taxa contributing to the metabolic pathways of the studied nutrient cycles is in accordance with the premise that soil and sediment microbial communities present elevated functional versatility (Barnes, Carter & Lewis, 2020), where many taxonomic groups can play considerable roles in a variety of functions.

We observed that the potential functional abundance for the metabolism of specific elements were spread across taxa as a function of biodiversity. Specifically, the impacted site, with a higher number of taxonomic groups, had fewer taxa making large contributions (>5%) to the metabolic potentials, while the conserved site with low diversity showed a higher number of families contributing greatly to such metabolisms. Some taxa, such as Desulfobacterales, an important sulfate-reducing bacteria, have been previously shown to contribute greatly with important nutrient metabolisms in mangrove sediments (Ullah et al., 2017; Nie et al., 2021). This is in accordance with our results where the Desulfobacterales taxa was also an important contributor to carbon and nitrogen metabolisms in both mangrove areas (Fig. 7). Taken together the results support the conclusion that, the lower the biodiversity, the higher the importance of particular groups to the nutrient cycles.

Considering that the conserved site shows that a large portion of the ecosystem function is dependent on a small number of taxa and that large differences exist between the taxa of the two sites, it may be expected that the urbanization process and the subsequent disturbance in the microbiome population would lead to an imbalance in nutrient cycling, potentially impairing some important metabolisms. The analysis of functional profiles suggests, however, the opposite trend where the functional potentials for these metabolisms are higher in the impacted mangrove site. Despite this increased metabolic activity at the impacted site, the ultimate effect these changes have on metabolic output, such as carbon sequestration or methane production, is difficult to predict. Previous work has shown that the increased consumption of carbon may not lead to increased sequestration as increased decomposition may lead to lower carbon burial rates (Bulseco et al., 2019).

Furthermore, the potential maintenance of the KOs and modules that make up nutrient cycles is not a guarantee of a healthy ecosystem. Changes in nutrient availability caused by human interference have previously been shown to affect the diversity of sulfate-reducing (SRB) and sulfur-oxidizing bacteria (SOB) (Meyer et al., 2016), thus interfering in the cycling of this element. Indeed, we see numerous prokaryotic species negatively affected by human impacts and there is great potential for eukaryotic species to be impacted as well. The alteration in microbiome nutrient cycling driven by urbanization is potentially hazardous as it can increase primary production, leading to a higher spatial and temporal hypoxia in the aquatic environment (Houser & Richardson, 2010), resulting in a cascading effect in the system. Although Camamu Bay is a well conserved coastal system, it does contain low abundances of toxic cyanobacteria, such as Microcystis, that have the potential to form blooms in eutrophic waters, which, in turn, is consistent with habitat degradation trends reported in estuaries worldwide (Affe et al., 2018).

In this work we found widespread variance in taxa abundance between sites up to and including the possible extinction of families at the impacted site (Fig. 4). Although the majority of taxa observed at the conserved site persisted in the impacted site, some families could be identified only in the conserved sediments. Here, we consider the absence of groups in the impacted site as indicative of a possible local extinction caused by human interference. The absences of some families are consistent with local extinction driven by anthropogenic change, such as subgroup 9 of Acidobacteria, which despite having a relatively high abundance at the conserved site is entirely absent from the impacted site. Notably many members of this taxa are known to be oligotrophic with a strong negative correlation with the organic carbon in the environment (Kielak et al., 2016). Conversely, for some families, such as Caldilineaceae and Rubritaleaceae, whose ecological aspects are still not well documented, their absence in the impacted mangrove suggests that these groups could also be oligotrophic or sensitive to some components of urban pollution, warranting further study.

Similarly, some of the taxa that were enriched or unique for the impacted mangrove area could be directly correlated with the discharge of domestic sewage in the area. The prevalence of the family Pasteurellaceae is an example, since most of its subgroups are known pathogens of vertebrates and are not usually found outside hosts (Christensen et al., 2014). Additionally, many genera belonging to the family Spirochaetaceae are known for causing a variety of human diseases such as syphilis, Lyme disease, leptospirosis, and periodontal disease, among others (Karami et al., 2014). Furthermore, members of the Ruminococcaceae are among the most abundant groups found in the mammalian gut environment (Biddle et al., 2013). Veillonellaceae is a diverse family with varying degrees of antimicrobial resistance and is associated with disease in both animals and humans (Marchandin & Jumas-Bilak, 2014). Finally, the enrichment in the Rikenellaceae family could be further evidence of fecal contamination, as the presence of subgroups of Bacteroidales has been proposed as a predictor of the risk of waterborne diseases (Schriewer et al., 2010).

In addition to the concerns about the prevalence of known human and animal pathogenic groups, recent studies have also shown the dissemination of antibiotic resistance genes in diverse mangrove compartments, including the sediment-root continuum, with antibiotic resistance sorting independently of the associate microbiota (Wang et al., 2021; Imchen & Kumavath, 2021). Antibiotic resistance itself is highly correlated with anthropogenic interference in mangroves, which is, in turn, strongly determined by local socioeconomic factors (Imchen & Kumavath, 2021). In this sense, the anthropogenic impacts in mangrove ecosystems can lead to consequences beyond the ecosystem threats and health risks for the local human population, as it can potentially increase global health threats with the spread of antibiotic resistance in pathogenic microbes.

We also observed that the variance in communities between replicates was greater in the conserved samples. While this could represent higher diversity in conserved microbiomes, other factors could potentially play a role. Importantly, sample collection at the conserved site spanned a larger distance with 15 m instead of 5 m separating the sample sites, and there was a more pronounced separation of tidal zones as well. The small number of samples is also a relevant limitation, especially for matters of statistical analysis. Additionally, the two sites, although sharing the same watershed and estuary, were ∼10 Km apart, and as such, numerous environmental differences may contribute to the variance as well. Nevertheless, the largest difference observed in physical parameters was the presence/absence of urbanization at the site. Considering that the samples from the impacted area were collected near domestic runoff sources, while the conserved site presented no signs of anthropogenic impact, and that some of the taxa uniquely found in the impacted sediments are recognized as pathogenic and gut microbiome colonizers, we consider that these important differences are correlated with the observed human impacts.

Our results show that the microbial communities of sediments can be largely affected by land use in mangrove adjacent areas, and that the observation of some particular prokaryotic groups are potential bioindicators of such human impacts, as well as posing risks to the surrounding populations. Notably, Brazilian mangroves are currently under elevated risk due to recent changes in the environmental legislation in the nation. Therefore, decision making organizations should be aware of the sensitivity of the microbiomes in this ecosystem, which play such important roles in the larger biogeochemical cycles, thus affecting global nutrient cycling, and act towards the conservation of the mangroves.

Conclusions

Our study looked at the effects of urbanization on the prokaryotic communities of mangrove sediments, through a comparison of samples taken from both impacted and conserved areas of the same estuarine system. The analysis found a statistically relevant change in the structure of the communities and an elevation of prokaryotic biodiversity in the sediments of the urbanized mangrove area. The predicted functional analysis showed that the general patterns of nutrient metabolisms could be maintained and that the metabolic potentials in the impacted mangrove were higher than in conserved mangroves, likely due to the higher biodiversity present in the impacted area. The observation of diverse groups contributing to metabolic pathways suggests higher versatility in the impacted mangrove sediments, where the functions are carried out by a greater number of groups, while fewer groups are responsible for large functional contributions in the conserved site. Beyond the differing nature of the sources of organic compounds and clear physical impacts between the areas, other environmental factors also played a significant role in the structure of the microbial communities. Notably, some of these may act synergistically to create the different patterns observed. Despite increasing biodiversity on the mangrove ecosystem, the presence of an urban settlement showed some clear negative effects, such as the extinction of some prokaryotic groups, as well as the colonization by human and animal pathogens in the microbiome of the impacted area. In this sense, the relative stability observed in functional terms does not imply the absence of a negative impact and recent studies confirm that human interference can lead to the spread of antibiotic resistance in mangrove sediments, which can cause serious health issues globally. This study provides evidence that the impacts of urbanization are reflected also in microbiological scales inducing important changes in ecosystem functions that could, in turn, impact the biogeochemical cycles in larger scales if not monitored and controlled. Future microbiome studies should be expanded to include eukaryotes, fungi, and viruses for a more complete profile of the microbial populations in these areas. Direct measurements of carbon, contaminants, and nutrient cycling rates in the field would also be valuable to verify the functional profiling results obtained in this work.

Supplemental Information

Supplemental Information 1 Full sized figure of collection spot in the impacted mangrove site

Figure shows clear signs of human disturbances in the mangrove. We emphasize the pipe coming from inside the construction, discharging domestic sewage directly in the mangrove sediments.

Click here for additional data file.

Supplemental Information 2 Supplemental Methods, Figures, and Tables

Click here for additional data file.

Supplemental Information 3 Abundances of ASVs in the sediment samples

Click here for additional data file.

Supplemental Information 4 Taxonomic abundances

The biom file with taxonomic abundances per sample used in the analysis.

Click here for additional data file.

Supplemental Information 5 Median NSTI scores for Taxa at the family level

PICRUSt2 scores taxa using the nearest sequenced taxon index (NSTI). This records a distance between the query 16S rRNA and the reference. This table contains the median and standard deviation of all queries assigned to a family.

Click here for additional data file.

The authors would like to thank the Organização de Conservação da Terra (OCT) for providing the structure for the field work in the environmental protection area, and the sequencing facility of the Microbial Ecology and Biotechnology Laboratory (LEMBiotec).

Additional Information and Declarations

Competing Interests

Author Contributions

Field Study Permissions

Data Availability

The authors declare there are no competing interests.

Carolina O. De Santana conceived and designed the experiments, performed the experiments, analyzed the data, prepared figures and/or tables, authored or reviewed drafts of the paper, and approved the final draft.

Pieter Spealman analyzed the data, prepared figures and/or tables, authored or reviewed drafts of the paper, and approved the final draft.

Vania Melo, David Gresham and Eddy Oliveira analyzed the data, authored or reviewed drafts of the paper, and approved the final draft.

Taise de Jesus and Fabio Alexandre Chinalia conceived and designed the experiments, performed the experiments, authored or reviewed drafts of the paper, and approved the final draft.

The following information was supplied relating to field study approvals (i.e., approving body and any reference numbers):

All sampling sites were conducted on publicly owned land, being either in the park proper or in designated public spaces within Ituberá City. We did rely on, and coordinate with Rogério Ribeiro, the Environmental Conservation Leader at Organização de Conservação da Terra (OCT) which manages conservation actions within the protection area, to guide our site selection and sampling.

The following information was supplied regarding data availability:

The computational workflow is available at

GitHub: https://github.com/pspealman/Project_Impact.

The impacted mangrove sediment sequencing data is available from NCBI BioProject PRJNA650560, and the pristine mangrove sediment data is available from NCBI BioProject PRJNA608697.

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
