# Peer review of "Large-scale differences in diversity and functional adaptations of prokaryotic communities from conserved and anthropogenically impacted mangrove sediments in a tropical estuary"

_PeerJ, doi:10.7717/peerj.12229_

## Round 0.1 · original submission · Major Revisions

Please provide a detailed point-by-point rebuttal letter to each of the reviewers' comments, along with your revised manuscript.

Reviewer 1 ·

Basic reporting

The major reported findings includes: 1) the environmental factors of organic matter, pH, salinity, dissolved oxygen and lead were significantly correlated with the observed shifts in the communities; 2) contributions of specific taxa to the functional potentials was negatively correlated with biodiversity, such that fewer number of taxa in the conserved area contributed to the majority of the metabolic potential; 3) the contamination by domestic sewage generated a eutrophic environment that may have led to the extinction of taxa observed at the conserved site.

Experimental design

Good. more detailed information reference the "General comments for the author".

Validity of the findings

Good. more detailed information reference the "General comments for the author".

Additional comments

In this MS, the authors focused on the 16S rRNA gene sequencing to investigate the difference communities between impacted and conserved in mangrove sediment. The major reported findings includes: 1) the environmental factors of organic matter, pH, salinity, dissolved oxygen and lead were significantly correlated with the observed shifts in the communities; 2) contributions of specific taxa to the functional potentials was negatively correlated with biodiversity, such that fewer number of taxa in the conserved area contributed to the majority of the metabolic potential; 3) the contamination by domestic sewage generated a eutrophic environment that may have led to the extinction of taxa observed at the conserved site. The weakness as described below, which should be improved upon before acceptance.

More specific notes below:

Line 49: The reference format in the text should be unified. Checking the reference format in the full text.

Line 101-102: Coordinate symbols should be unified

Line 112:What is the material of the cylindrical sampler mentioned, and does it have an effect on environmental factors of the mangrove sediment such as heavy metals?

Line 121: provide what instruments and method were used for analyzing the environmental factors of heavy metal.

Line 154: “(S02 Fig.,)”, check this sentence.

Line 154-159: “SILVA database (Silva SSU 132), (McDonald et al. 2012), (see Supplemental Methods).” check journal style of language in the full text.

Line 160: “… (Dixon 2003)in …”, check this sentence.

Line 163, check this sentence.

Line 218: It is mentioned that fig2B is a NMDS plot, and the annotations are a PCoA plot. If it is a NMDS plot, please provide the stress value.

Line 220: The statistically significant of p values required less than 0.05, checke the sentence “We found significant (p < 0.1) differences between the population
structures of the two communities, suggesting that the environmental effects of the urbanization on the mangrove has generated a significantly different microbiome.”

Line 236-237: “Many of the families that were abundant in the impacted mangrove site were absent in the sediments of the conserved area.” Rewrite this to provide some details.

Line 285: What is the typical range of salinity in mangrove sediment? Will the salinity variation be significant and expected to impact the bioavailability in the study?

Line 302-309: How do these environmental factors (dissolved oxygen, pH …) affect the microbial community. Rewrite this to provide some details.

Line 302-309: “This is in accordance with our results where the Desulfobacterales taxa was also an important contributor to carbon and nitrogen metabolisms in both mangrove areas.” refer this sentence to the figure/table.

Reviewer 2 ·

Basic reporting

The standard of English is good and generally well structured. Raw data has been provided.

Some detail is missing from the introduction and discussion (see general comments for more information) and the references provided are not always the most suitable (or relevant) to the associated statement.

Experimental design

The study contains original primary research within the aims and scope of the journal.

The reasoning behind the need to fill the knowledge gap is sound.

A lot of important details are missing from the methods section.

The hypothesis or hypotheses need to be adjusted to match what was actually measured in this study.

The replication of samples is very low which does not give the reader strong confidence that the study questions have been adequately answered.

Validity of the findings

Conclusions are well stated but the authors need to give more consideration in the discussion section to the limitations of their study, particularly in relation to level of replication and the methods used to asses sediment function.

Additional comments

General comments
Please be consistent throughout in how the sites are named, sometimes it is “preserved” and other times it is “conserved”.

It needs to be made clearer in all sections that the prokaryotic functional profiles are estimated from the prokaryotic taxonomy and the limitations of this method need to be fully considered in the discussion section.

Introduction
Line 49: Mangroves do not account for more than 70% of the world’s coastline. Please recheck the literature. Of the papers that you reference here, none seem to be relevant/appropriate for this statement: Yao et al 2020 is a study on mangrove bacterial epiphytes, Singh et al 2005 is a review on mangrove nutrient cycling, and Imchen et al 2018 looks at metagenomes for effects of heavy metals an antibiotic resistance. I suggest you seek out some of the many papers that actually measure or review mangrove area cover.

Also multiple references at the same point should be in the same brackets, not in separate adjacent brackets.

Line 50-51:Again, the references provided here are perhaps not the most appropriate. Find actual studies or reviews that provide mangrove areas for different countries.

Line 53: Again, Moitinho et al 2018, referenced here, is not relevant to the statement at hand. Just because they provide a summary of mangroves in their opening paragraphs, they are not the source of this information. Seek out papers that actually assess carbon sequestration capacity and coastal protection.

So far nearly all of the referenced papers are not particularly suitable as supporting evidence for the accompanying statements. I am not going to check every single reference in the paper to see if it is relevant to the statement. I highly recommend that during the revision the authors take the opportunity to go through and double check that the references they provide are actually relevant to the accompanying statements.

Line 86-88: this study doesn’t really provide understanding of the consequences of urbanisation in the larger ecosystem. It is evaluating local effects near the impact. Consider revising this statement.

Line 89-91: “We hypothesized that the constant input of nutrients present even in only light urban domestic runoff and sewage would lead to a more eutrophic environment in the mangrove sediments close to the urban area.” This is not the hypothesis of the study, nutrients were not even measured. Please adjust the wording of this paragraph to relate your hypothesis to the thing you are actually testing (that proximity to urban environments will increase/decrease/change prokaryotic diversity/composition/function/other)

Line 91-94: “The higher diversity and bioavailability of nutrients favors the colonization of exogenous opportunistic prokaryotes while leading to the extinction of more sensitive endogenous taxa, ultimately creating significantly distinct communities with distinct metabolic profiles.” The authors have not provided any information in the introduction to support this statement. Please provide more information in the introduction summarising what has already been found in studies that have assessed effects of sewage/increased nutrients on mangrove sediment prokaryote community composition/diversity/function. This should inform your hypotheses for what you expect to find.

Please provide justification for why only prokaryotes were assessed in this study and why they are important in the functioning of mangrove sediments.

Line 96: “microbial communities” should be changed to “prokaryotic communities” as the whole microbial community was not evaluated in this study.

Methods & Materials
Line 101-102: saying “located ..between the coordinates...” and then only listing one latitude coordinate and one longitude coordinate does not make sense. You either need to give two sets of coordinates, or reword to say the coordinates are for the centre of the area, or instead just provide two sets of coordinates – one for each of your sampling locations.

Line 112: I do not think “superficial” is the word the authors meant to use here, should this be surficial?
Please provide much more information relating to the sampling sites, sampling locations, and sampling methods. How far are the two sites from each other? How far apart were the replicate samples collected at each site? Where in the tidal zone were the samples collected – was it similar at each site? Were they collected at high or low tide? When were they collected – month/year/season? What mangrove tree species are present at each site? How far from trees were cores collected? Was the sediment type similar at each site – sand/silt/mud/mixed? Were the samples stored and transported on ice to the laboratory? It says 3 composite samples per site – how many cores were combined for each composite sample? What kind of area was covered by the sampling? What was the diameter of the sampling core? Were the samples collected along a transect? From plots? How were sampling locations determined? How far from habitation were the impacted samples located?
This section does not mention how the heavy metals were analysed.
Is says D.O., conductivity, pH and temperature were measured in the water column – how far were these measurements taken from the collected sediment cores? Directly above? Was the sediment underwater at the time of sampling?

Line 128: says amplification of the bacterial 16S rRNA gene but you also assess archaea from this data so “bacterial” should be removed here.

Were any negative or positive controls included in the PCRs and were they sequenced?

What was the range of DNA concentrations of the extracts? It says 2.5 uL of each sample was added to the PCR reactions but that gives no indication of how much DNA was added.

What cleanup method was used after PCR?

How were libraries quantified and pooled?

Line 153-155: was dataset normalised (eg. rarefied) before diversity and community structure comparisons? I note that sequencing depth was lower in the pristine samples (Fig S1) and so therefore without normalisation would already be expected to have lower diversity than impacted samples which is not necessarily wholly related to the sediment samples themselves. I see that the authors have rarefied data prior to differential abundance analyses (lines 175-177) even though it is arguably not considered the best normalisation to use for this type of analysis (eg. McMurdie & Holmes 2014).
Was there any trimming of the dataset based on numbers of reads? For example were ASVs kept in the dataset even if they only had a total of 1 read across all samples?

Line 165: It would be useful to start this sentence saying what PICRUSt was used to assess ie. to predict the functional profiles of the prokaryotic communities based on their taxonomic composition

It might be helpful to split the Data analysis section up into the bioinformatics processing of the sequencing data, and then statsistical analyses performed. There appears to be information missing relating to statistical analyses. What statistical tests were used to compare diversity measures between sites? ANOVA? What test was used to assess community composition between the areas? Were the data checked for normality and other assumptions? The results section refers to a permanova but this is not mentioned in the methods. The results figures include an NMDS and a PCoA but these are not specifically mentioned in the methods.

Results
I think it is important to know what differences there are in environmental variables between the two sites you are comparing to understand why there are differences in prokaryotic communities. Instead of presenting this in the supplementary information, I suggest that it is included in the results and that differences between the sites are tested statistically.

Line 204: I would be cautious about stating that the communities are mainly composed of bacteria as sequences reads do not necessarily equal abundance. qPCRs with bacteria and archaea specific primers would be required for such a comparison. I would suggest rewording to say that the ASVs in the dataset were mainly composed of bacteria.

Line 204-206: In this sentences when it is referring to number of sequences it is not entirely clear whether the authors are referring to the number of unique sequences (ie. ASVs) or the number of sequence reads. I am assuming the latter but this should be clarified. It would also be useful if the authors could include how many ASVs were in the entire dataset, how many ASVs were detected at each site, and how many ASVs overlapped between the two sites. It should also be made clear whether the numbers provided are from normalised or non-normalised data.

Line 216-220: I would suggest rewording of this sentence to improve readability and accuracy. It makes it sound like the NMDS plot is calculated using a permanova test which is not correct. It would make more sense to present the permanova results when mentioning the permanova test. Also it is probably not necessary to include all the additional methodology wording here that is already presented in the Methods section and the figure caption. Suggested alternative: “Prokaryotic community composition differed between the two sites (insert permanova results) with the samples of each mangrove site forming separate clusters on the NMDS plot (Figure 2B).”

I would prefer to see more detailed stats outputs eg. the degrees of freedom, F, R2 and p values so that the reader can assess for themselves how strong the relationships are. Is the p value presented in line 220 (p<0.1) the results of the permanova? If so, this would not normally be considered significant and should be reworded to marginally significant.

Why is there only one impacted point on the NMDS plot? Is it because the two sites are too different to present on the same plot and the impacted ones are all overlapping because they are less variable? This needs some kind of explanation as this is not a normal NMDS plot. Nowhere in the results is it mentioned that the composition is more variable at the non-impacted site, this is an important point to note and consideration should be given to why. I see that the authors also present a PCoA in Figure 5 which effectively renders the NMDS plot redundant. I would suggest removing the NMDS plot and refer to the PCoA instead of the NMDS when talking about differences in community composition.

I can’t seem to see anywhere in the results where the authors refer to the result presented in Figure 2C. This should be included in the text or the figure removed.

In this same paragraph when presenting the differences in community composition, it would make more sense to present the environmental variables that relate to the community composition instead of coming back to that later on in the results section.

Line 221-222: “suggesting that the environmental effects of the urbanization on the mangrove has generated a significantly different microbiome.” This should be kept for the discussion section rather than the results. I also do not think that this is necessarily supported by the results presented, considering there is only one impacted and one non impacted site with low sample numbers in different locations of the estuary.

Line 223: “Alpha-diversity measured were performed...” diversity measured are not performed, they are not tested. The measures were calculated. Also please include more of the stats output than just the p value eg. “ Shannons diversity index was significantly higher in the impacted sediments compared to the non-impacted sediments (Kruskal-Wallis chi-squared =...., df = ....., p-value = ... )”. Also Figure 3 presents Simpsons diversity but this is not mentioned in the text. Either include in text or remove from Figure 3 to supplementary.

It would be good if the authors could also present the richness (ie. number of ASVs) results here too. In the text and in Figure 3.

Lines 228-240: Much of the text in this paragraph belongs in the methods or the discussion section rather than the results. For example: “Biotic and abiotic selective pressure at the impacted site can alter the relative fitness, and therefore abundance, of taxa. At its most extreme this can lead to native taxa becoming locally extinct and allow for the colonization of exogenous prokaryotes” this is discussion not results. “In order to identify possible opportunistic and extinct taxa in the two mangrove sediments, we identified families with significant (p < 0.05) differences in abundances between sites” this is methods not results. “These results suggest that some taxa present in the conserved site, here representing the most natural conditions, adapted well to the changes at the impacted site, while other groups struggled in the modified environment, being led to extinction” this is discussion not results.
Perhaps provide more info from this analysis like how many taxa were identified as being differentially abundant? How many of these taxa were only found in the impacted and how many were only found in the non-impacted area? Are any of these families known to have important functions in sediments?

Line 258-262: “Importantly, Formaldehyde is a common organic compound found in animal waste, building materials, pesticides, and fertilizers. While formaldehyde assimilation is a common intermediary in the oxidation of methane by methanotrophs, it is also known that the influx can alter microbiomes (Dixon 2003), suggesting that the prokaryotes at this site may have been selected to take advantage of this additional carbon source present in the urban runoff” this is discussion not results.

Discussion
Once the authors have adjusted their hypotheses in the introduction, they should then come back to this in the discussion in relation to how their findings do or do not support the hypotheses.

The authors attribute increased prokaryotic diversity to eutrophication, but nutrients were not measured in this study, and nitrogen addition has mostly been found to lead to decreases in microbial diversity in terrestrial soils (see Wang et al 2018), with similar results observed in nutrient addition experiments with coastal soils (eg. Aoyagi et al 2015; Wang et al 2016; Bulseco et al 2019; Craig et al 2021). The authors should either not attribute the findings to something that wasn’t measured, or consider their findings in the context of other studies and suggest reasons why their results differed.

In relation to the discussion on salinity, although salinity did differ between the two sites, in biological terms the difference between a salinity of 13.3 and 15.1 is unlikely to have much impact.

Please provide discussions around the limitations of the study (eg. small sample size, no replication of sites, functional profiles estimated from taxonomic data, nutrients not measured). Conclusions have been drawn from a very small number of samples from only one “conserved” and on “impacted” site. Due to the different locations in the estuary, two such sites would be expected to have differences in their environmental conditions anyway and this paper provides no clear evidence that the differences in are the direct result of anthropogenic activity.

Perhaps include suggestions for future work that could be done to further assess affects on functioning and the implications. eg. Actual measures of carbon and nutrient cycling in the field need to verify the functional profiling results

The authors continually relate the results to sewage but as far as I can tell from the information provided, the samples weren’t collected specifically near a sewage outlet. Therefore could there be other potential anthropogenic impacts at play? Chemical pollutants? Physical disturbance?

What are the potential implications of the “higher functional abundance of the methane and carbon fixation pathways at the impacted site? Could this lead to changes in greenhouse gas emissions?

Line 400: I think “organs” is supposed to be organisations. Or bodies

Figures
Figure 2A: Each of these panels (impacted and conserved) had three samples. Are the results presented in this figure a sum of all the read from each of the three samples? Or the mean? Median? Please make this clear in the caption. Also, there is a grey colour that is not on the legend. I assume that it means they were unassigned at the class level but this should be specified.

Figure 2B: The figure caption says it is a PCoA plot but the figure axes and methods in the report say that it is an NMDS plot.

Figure 2C: Why does the x-axis say conserved n=9? should this not be n=3?

Figure 3: I would like to see richness (number of ASVs) added to this figure. Also, Simpson’s index is presented in the figure but not mentioned in the figure caption or in the text. And I don’t understand the difference between the plot on the left and the plot on the right. They both seem to be showing shannons index for impacted and conserved. But the numbers are different??? Is the plot on the left showing mean? and the box plot showing median with interquartile ranges? Even then the differences in numbers don’t make sense. More info needed in caption and I think one of these figures is incorrect. There is no need to present both types of figure so choose one and make sure that the numbers are correct.

Figure 4: This is a nice figure. The authors should extract some of the more important details to present in the results text too.

Supplementary files
Environmental variables: this file provides no units for the numbers in the table. Is it air or sediment temperature? Please specify what D.O. and O.M. abbreviations stand for. Which variables were measured in water and which were measured in sediment?

I can’t find titles or captions for any of the supplemental figures.

Figure S1: should the y-axis title be observed number of ASVs? It does not mention in the paper anywhere clustering of the ASVs into OTUs

Figure S3: y-axis text is too small to read. Also S3B x-axis has a typo “Conservedd”

Figure S4: I have no idea what this figure is showing. It either needs some explanation, or if it is of no use then remove it.

Figure S5: The y-axis title should read proportion not percent to match the numbering used on the scale. I am surprised that only 70-80% of taxa were unassigned at the species level, I typically get a much higher number.

References
McMurdie & Holmes 2014 Waste Not, Want Not: Why Rarefying Microbiome Data Is Inadmissible. https://doi.org/10.1371/journal.pcbi.1003531

Aoyagi et al 2015 Dynamic transition of chemolithotrophic sulfur-oxidizing bacteria in response to amendment with nitrate in deposited marine sediments. https://doi.org/10.3389/fmicb.2015.00426

Bulseco et al 2019 Nitrate addition stimulates microbial decomposition of organic matter in salt marsh sediments https://doi.org/10.1111/gcb.14726

Craig et al 2021 Nitrogen addition alters composition, diversity, and functioning of microbial communities in mangrove soils: an incubation experiment. https://doi.org/10.1016/j.soilbio.2020.108076

Wang et al 2016 Responses of bacterial and archaeal communities to nitrate stimulation after oil pollution in mangrove sediment revealed by Illumina sequencing. https://doi.org/10.1016/j.marpolbul.2016.05.068

Wang et al 2018 Decreasing soil microbial diversity is associated with decreasing microbial biomass under nitrogen addition https://doi.org/10.1016/j.soilbio.2018.02.003

---

## Round 0.2 · Minor Revisions

Thank you for your revised version. The manuscript requires a few more minor amendments. Please provide your point-by-point reply to them along with the revised manuscript.

Reviewer 1 ·

Basic reporting

The standard of English is good and generally well structured. Raw data has been provided.

Experimental design

The study contains original primary research within the aims and scope of the journal.
More detailed information reference the "General comments for the author".

Validity of the findings

Conclusions are well stated, linked to original research question.

Additional comments

The font size and the language style in the full text need to check. The full text requires a strict checking.
For example:
Line 49: “Introductio”. Spelling mistake.
Line 50: “… 137,760 km2 of the world’s coastline …”. Checking this sentence.
Line 162: “Briefly, 0.5g of the …”, and line 175: “… from 0.25g of each sediment …”. A space is required between letter and number.
Line 420: “We calculated the functionalFunctional profiles …”. Spelling mistake.
...

Reviewer 2 ·

Basic reporting

The language is good but some further clarification in parts of the sampling design is required (see additional comments). Background/context information and references is sufficient. Article is well structured, and raw data is available. The hypothesis is now better suited to the study.

Experimental design

Research question is well defined. The revised version now has much more methodological detail which is useful, there are just a couple of things that could be further clarified (see additional comments).

Validity of the findings

Although replicate numbers are low, and there are multiple additional factors that differed between how samples were collected and analysed between the two sites, the study does provide some evidence that mangrove sediment bacterial community composition is impacted by urbanisation/human factors.

Additional comments

• Have I interpreted correctly that one site was sampled in July 2018 and the other in February 2020? And different tidal zones were sampled at each site? And the samples for each site were sequenced on different sequencing platforms MiSeq vs NovaSeq with different read lengths (300 cycles vs 2x250 cycles). And the data for each site were processed with different settings for the denoising step. Each of these factors are introducing so many additional differences between the samples that how can you confidently attribute and bacterial community differences to the conserved vs impacted factor? I do concur though with the comments in L472-474 that “…some of the taxa uniquely found in the impacted sediments are recognized as pathogenic and gut microbiome colonizers, we consider that these important differences are correlated with the observed human impacts”. I think this point which is based on presence/absence is sufficient to show that urbanisation does have some impact on the community composition even if some of the other findings, which are reliant on differential abundance of sequence read numbers are a bit more questionable.
• With regards to the heavy metal analysis – was this conducted on the same sediment samples that were used for the DNA extractions? Or are you using existing data from a previous study on previous samples? If the latter, approximately how far away and when were the heavy metal samples located? It is not ideal to be using heavy metal data from different samples collected from different locations at different times for drawing conclusions of the effects on bacterial communities.
• L40 “…originally accounting for more than 137,760 km2 of the world’s coastline…” The word originally does not make sense in this sentence.
• L90 “…we calculated the general patterns of functional potentials in these microbiomes…” I would suggest rewording this to “we estimated” or “we calculated estimated functional potentials”
• L98: there appears to be a typo here with the letters “remo” after the reference
• L121-122 “The mangrove trees are composed of the species Rhizophora mangle, Avicennia schaueriana and Laguncularia racemosa” could you please clarify if these same three tree species were present in the vicinity of the sampling area at both sites?
• L133-134: it says that DNA was extracted from 0.25g of each core. The results indicate that there are a total of three samples for each site, but more than three cores are mentioned for each site in the sampling methods section. Please provide some clarity as to whether replicate cores were pooled prior to extraction (if so then this line should say DNA was extracted from 0.25g of each composite sample), or if the DNAs for each core were pooled after extraction. Or from the NCBI database it looks like there is sequencing data for 9 samples at the pristine site, 3 from each tidal zone, was the data pooled instead for the site? It does not seem to be the same for the other site which appears to have sequencing data for just three samples.
• L188: It is not understood what the authors mean by “Alpha-rarefaction was calculated…”. Are they referring to the production of rarefaction curves, which are presented in Article S1 Figure 2, or are they saying that they rarefied (normalised) the data prior to analysis of alpha- and beta-diversity? Rarefy and rarefaction are two different terms, one is a normalisation technique and one is a technique assessing species richness from the results of sampling
• L287-288 “The analysis revealed significantly higher richness and diversity (Shannon’s diversity, Kruskal-Wallis H = 3.857, df = 1, p-value = 0.050)…” Why are statistical results provided in this sentence for Shannon’s diversity but not richness?
• I note that the authors have added more information regarding the locations of the samples and the number of cores, but it is still a little unclear and requires rewording. For example (L108-111): “At the conserved mangrove site (13°44'34.6"S, 39°03'30.6"W), collection points were 15 meters distant from each other and 3 sediment samples of each tidal zone (supralittoral, intertidal and sublittoral), composed by 3 sediment cores randomly extracted, were collected in July 2018 (de Santana et al. 2021).” From what I understand there were three samples for this site – did each sample consist of one core from each zone mixed together? If so, were these collected along a transect perpendicular to the water? Or were the three samples each representative of one zone? And how do the other three randomly extracted cores fit in? Was each subsample core 15m apart? Or a transect for each sample were 15m apart from each other?
• Also, with the impacted site (L114) it is unclear if the subsample cores for each sample were 5m from each other, or if the areas in which cores were collected for each sample were 5m apart.
• Were the sublittoral samples within the mangrove tree area or were these in the water channel beyond the tree line?

---

## Round 0.3 · accepted · Accept

Thank you for addressing the reviewers comments and suggestions, your paper can now be accepted for publication in PeerJ.